# Dietary adherence among persons with type 2 diabetes: A concurrent mixed methods study

**Dorothy Wilson**[1]*, **Abigail Kusi-Amponsah Diji**[1], **Richard Marfo**[1], **Paulina Amoh**[1],
**Precious Adade Duodu**[2], **Samuel Akyirem**[3], **Douglas Gyamfi**[4], **Hayford Asare**[1],
**Jerry Armah**[5], **Nancy Innocentia Ebu Enyan**[6], **Joana Kyei-Dompim**[1]

**1** School of Nursing and Midwifery, College of Health Sciences, Kwame Nkrumah University of Science and Technology, Kumasi, Ghana, **2** Department of Nursing, School of Human and Health Sciences, University of Huddersfield, Huddersfield, England, United Kingdom, **3** School of Nursing, Yale University, West Haven, Connecticut, United States of America, **4** University of Maryland School of Nursing, Baltimore, MD, United States of America, **5** College of Nursing, University of Florida, Gainesville, Florida, United States of America, **6** School of Nursing and Midwifery, University of Cape Coast, Cape Coast, Ghana

* drwilsondorothy@gmail.com

## Abstract

### Background

Poor adherence to dietary recommendations among persons with type 2 diabetes (T2D) can lead to long-term complications with concomitant increases in healthcare costs and mortality rates. This study aimed to identify factors associated with dietary adherence and explore the barriers and facilitators to dietary adherence among persons with T2D.

### Methods

A concurrent mixed methods study was conducted in two hospitals in the Ashanti Region of Ghana. One hundred and forty-two (142) persons with T2D were consecutively sampled for the survey. Dietary adherence and diabetes-related nutritional knowledge (DRNK) were assessed using the Perceived Dietary Adherence Questionnaire (PDAQ) and an adapted form of the General Nutritional Knowledge Questionnaire (GNKQ-R) respectively. A purposive sample of fourteen participants was selected for interviews to explore the factors that influence dietary adherence. Qualitative data were analysed using NVivo version 20 software and presented as themes. Furthermore, binary logistic regression was performed using IBM SPSS version 29.0 to identify the factors associated with dietary adherence.

### Results

Nearly fifty-one percent (50.7%) of the participants in this study had good dietary adherence. In multivariable logistics regression, it was found that increase in DRNK (AOR = 1.099, 95% CI: 1.001–1.206, p = 0.041) score and living in an urban area (AOR = 3.041, 95% CI: 1.007–9.179, p = 0.047) were significantly associated with good dietary adherence. Inductive thematic analysis revealed four facilitators of dietary adherence (access to information on diet, individual food preferences and eating habits, perceived benefits of dietary adherence, and presence of social support) and four barriers (inability to afford recommended diets, barriers related to foods available in the environment, conflict between dietary recommendations and individual eating habits, and barriers related to the social environment).

**Funding:** The author(s) received no specific funding for this work.

**Competing interests:** The authors have declared that no competing interests exist.

## Conclusion

The findings support the need for interventions including continuous dietary education tailored to individual preferences and dietary habits, expansion of poverty reduction social interventions and formulation of policies that will improve access to healthy foods in communities.

## Introduction

Despite the significant efforts made to prevent and control diabetes [1–3], the disease remains a global crisis with increasing prevalence in all regions of the world. According to the International Diabetes Federation (IDF) [4], about 537 million adults aged 20–75 years globally (10.5% of all adults in this age group) were living with diabetes as of 2021. This number is expected to rise to about 783 million (representing a worldwide prevalence of 12.2%) by 2045 with 94% of the increase occurring in low and middle-income countries. In the IDF Africa region, about 24 million adults within the same age group were living with diabetes, representing a regional prevalence of 4.5% [4]. Additionally, about 6.7 million deaths occurred in the world due to diabetes in 2021 [4].

Type 2 diabetes (T2D) is the most common type of diabetes that accounts for over 90% of all diabetes worldwide [4]. The growth in the worldwide prevalence and incidence of T2D is driven by rapid economic and social development resulting in a change in dietary patterns and decreased physical activity [5]. The management of T2D is complex and requires more than pharmacologic treatments. Self-management practices including eating a healthy diet, physical exercise, weight control, and smoking cessation are the foundation of the management of T2D [4].

Dietary management of T2D is of crucial importance in achieving glycaemic control and preventing the development of long-term complications such as cardiovascular disease, kidney disease, visual problems, and neuropathy [6]. Diets high in whole grains, fruits, vegetables, legumes, and nuts, as well as moderate alcohol use and low in refined grains, red/processed meats, and sugar-sweetened beverages, have been shown to minimize diabetes risk and improve glycaemic control and blood lipids in persons with T2D [7]. Despite the important role diet plays in the management of T2D, the prevalence of non-adherence to dietary recommendations is high in both developed and developing countries [8–10]. Possible influencing factors that have been cited in most literature include lack of knowledge, sociodemographic characteristics, the presence of comorbidities, and family and social support systems among others [8, 11–14]. Understanding the factors that influence dietary adherence among persons with T2D is important to enable stakeholders to design and implement interventional programs to promote adherence and subsequently lead to improved health outcomes in persons with T2D. Due to this, several quantitative and qualitative studies have been conducted in developed countries and some sub-Saharan African countries on dietary adherence [8–10, 13, 15–19]. However, findings from these studies may not apply to the Ghanaian population due to differences in sociodemographic characteristics, sociocultural practices, and dietary patterns in the different settings. From our literature review, most of the studies conducted on self-management of T2D in Ghana employed quantitative methods [14, 20–22] to identify factors associated with adherence to self-management practices with only a few studies [23] qualitatively exploring the challenges that persons with T2D face and the factors that enable adherence. Considering the quantitative findings, level of education or years of education; marriage;

social support systems, socioeconomic status and other factors such as illness perception and diabetes knowledge have been found to be associated with dietary practice and other diabetes self-management practices [14, 20–22]. Meanwhile, the few qualitative studies in Ghana have reported that the social and physical environment, economic situation, and individual preferences may constitute either facilitators or barriers to dietary adherence. Barriers to dietary adherence include inadequate family support, the temptation to eat less healthy foods at social events, cultural beliefs about weight, difficulty changing old habits, poor income levels, difficulty accessing a variety of foods, and dislike of recommended diets [23, 24]. On the other hand, the facilitators of dietary adherence include support from family and healthcare professionals and the patient's hopes for a cure [23].

Considering the flaws inherent in the quantitative or qualitative methods, the use of study designs that triangulate the findings is required to provide a better insight into the issue. Using both qualitative and quantitative methods offer the advantage of maximising the strengths and minimizing the flaws of each approach [25]. For example, Han et al [26], conducted a mixed methods study to understand the relationship between diabetes-related nutritional knowledge (DRNK) and dietary adherence among persons with T2D in Singapore. Surveys were used to collect data on DRNK, and dietary adherence and statistical analysis were performed to determine the correlation between them. Meanwhile, semi-structured interviews were conducted to gain insight into the factors participants perceived as barriers and facilitators to dietary adherence. The statistical tests showed no correlation between DRNK and dietary adherence. However, the data from the interviews provided information on enablers and barriers to dietary adherence that possibly mediate the translation of DRNK into practice.

By virtue of the above-stated factors, the current study sought to identify the factors associated with dietary adherence, as well as explore the barriers and facilitators of dietary adherence among persons with T2D.

## Materials and methods

### Study design

The study employed a concurrent mixed methods design [27], whereby quantitative and qualitative data were collected and analysed within the same timeframe. The quantitative phase employed an analytical cross-sectional design, which involved collection of data using a structured questionnaire. The qualitative phase, on the other hand, employed a descriptive phenomenological design which involved individual interviews with participants.

The use of multiple methods or data sources can enhance the validity of the study findings by identifying biases that may be present in a single method and allowing the researchers to compare quantitative and qualitative results [28]. Moreover, it helps the researchers to gain a comprehensive understanding of the phenomenon [28]. Qualitative methods can provide rich description and context but may not be representative of a larger population, while quantitative methods provide statistical information which can be generalized to a larger population but may lack contextual information [29]. Additionally, quantitative data may be affected by errors in measurement [30] while qualitative data may be influenced by the subjectivity of researchers [29]. By employing multiple methods, the researchers can reduce the impact of these biases.

### Study setting

The outpatient diabetes clinics of two district-level hospitals in the Ashanti Region of Ghana were used as the study sites. One of the hospitals, the University Hospital, Kwame Nkrumah University of Science and Technology (also known as KNUST Hospital) is in the Oforikrom

Municipal, which is an urban municipality [31]. It is a quasi-government hospital and caters for staff of the Kwame Nkrumah University of Science and Technology (KNUST) and their dependants, students at the university and people from surrounding communities. The hospital has a 125-bed capacity and offers general medical services as well as specialist services.

The other hospital, Ejisu Government Hospital is a public hospital in the Ejisu Municipal, which constitutes both rural and urban localities [31]. It is a 95-bed capacity facility and offers both general and specialist services. Both hospitals hold outpatient clinics once every week for persons with diabetes. Persons who visit the clinic go through blood glucose and blood pressure monitoring, consultations by physicians and dietitians, and receive dietary advice in the form of general health education. Data was collected from persons with T2D who attended the diabetes clinics for one month (between September and October 2022).

The study sought to target both rural and urban population. Hence, the choice of the KNUST Hospital and Ejisu Government Hospital is based on their diverse patient populations. The KNUST Hospital provides services to a mix of students, university staff and locals while the Ejisu Government Hospital serves both rural and urban communities. The presence of patients from different socioeconomic backgrounds makes the two hospitals ideal settings to capture a wide range of perspectives on dietary self-care in diabetes.

### Study population

All persons with T2D who attended the outpatient diabetes clinics in the hospitals constituted the population for the study. The total population of persons with T2D who had been diagnosed for at least one year was estimated at 120 and 80 for KNUST Hospital and Ejisu Government Hospital respectively, resulting in a total of 200. The average monthly attendance was used for this estimation, and it was obtained from the outpatient diabetes clinic registers of the two hospitals.

### Inclusion and exclusion criteria

Participants who were eligible to be sampled for the study included those who: a) were 18 years and above, b) had a confirmed diagnosis of T2D for at least one year and had sought care from the diabetes clinic at least twice during the last 12 months, and c) could speak, understand, or write either English or Twi or both. Persons who had been diagnosed with gestational diabetes and were critically ill or had significant cognitive impairments excluded from this study.

### Sample size determination and sampling for the quantitative phase

The sample size for the quantitative phase was calculated using the Taro Yamane formula: $n = \frac{N}{1+N(e)^2}$ [32]. Where **n** signifies the sample size, **N** signifies the population under study, and **e** signifies the margin of error (0.05). The total population (N) was estimated at 200 using the average monthly attendance at both hospitals and the minimum required sample size obtained was 133. To compensate for registration errors and non-response, a contingency sample of 10% was considered, resulting in an estimated sample size of 146.

Using GPower [33] for a post-hoc power analysis, we determined that a sample size of 146 will yield >99% power at 95% confidence level. This determination was based on an assumed odds ratio of 4.7 between diabetes knowledge and dietary adherence as reported in a previous study in Sudan [34] and a two-tailed z-test.

Consecutive sampling techniques were used in selecting eligible participants for this study phase. Thus, all participants who met the inclusion criteria were invited to participate until the desired sample size was reached.

## Sample size determination and sampling for the qualitative phase

Participants for the qualitative aspect of the study were purposively sampled using maximum variation techniques. The researchers selected participants who differ across multiple criteria such as age, sex, place of residence, income, education level, employment, and marital status. The aim of employing this technique was to capture a variety of experiences and viewpoints. These participants were selected from the sample members recruited for the quantitative phase of the study. This is a method of integration known as **connecting** and it occurs when one dataset links to the other through the sampling frame [35]. Participants were invited to participate in the interview immediately after the quantitative data collection. Thus, participant selection and data collection for both quantitative and qualitative phases were done concurrently.

Data saturation (when new data replicates what has already been gathered) was achieved after conducting 14 interviews [36].

## Data collection tools for the quantitative phase

A structured questionnaire consisting of three sections was used for the survey. The first section had sociodemographic and clinical variables including age, sex, marital status, place of residence, educational status, monthly income, and employment status) and clinical variables status, (duration of condition, family history of diabetes, and presence of comorbidities). For the place of residence, the researchers obtained name of the locality of residence of the participants and designated them as rural or urban. All localities within the Oforikrom Municipality were designated as urban. For the Ejisu municipal, all localities apart from Ejisu (Krapa inclusive), Kwamo and Fumesua were designated as rural [37, 38].The second section had the Perceived Dietary Adherence Questionnaire (PDAQ) for the measurement of dietary adherence (outcome variable). PDAQ is a nine-item questionnaire that was developed and validated by Ghada Asaad et al. in 2015 [39]. The response is based on a seven-point Likert scale. Some of the questions on the scale include "On how many of the last 7 days did you eat foods high in sugar, such as cakes, cookies, desserts, candies, etc.?", and "On how many of the last 7 days did you space carbohydrates evenly?" Higher scores reflect higher adherence except for items 4 and 9, which reflect unhealthy choices (foods high in sugar or fat). To calculate a total PDAQ score, the scores for items 4 and 9 were inverted. The first and second items on the scale were changed to: "On how many of the last 7 days have you followed dietary recommendations given by your healthcare provider?" and, "On how many of the last 7 days did you eat the number of fruit and vegetables you are supposed to eat based on recommendations from your healthcare provider?" respectively. These questions were modified because, in the original questionnaire, they were based on Canada's Food Guide which may not apply in our setting. Additionally, the food examples given in item 3 and item 8 were modified to suit the context of the study area and study population. Participants were classified as having good dietary adherence if they obtained a score equal to or above the median score. The median score was used in this study because the Shapiro-Wilk test indicated that was not normally distributed (W (142) = 0.636, p<0.001).

The third section had nine questions from the revised version of the General Nutritional Knowledge questionnaire (GNKQ-R) [40] to assess participants' diabetes-related nutrition knowledge (DRNK). The questions were based on the following areas: number of fruits and vegetables eaten per day, oils to cut down, foods to cut down a lot or increase intake, foods high or low in fibre, foods high or low in starch, foods high or low in added sugar, foods with high glycaemic index, foods likely to raise blood cholesterol and healthy food choices. Some modifications were made to the food choices for a better understanding of participants, considering the local context while maintaining the relevance of each question in the original

questionnaire. Each item carried one point for a correct answer. Some questions had subsections and those sections were treated as separate items. The total possible score for each participant ranged from 0 to 27. A score equal to or above the median score was categorized as high DRNK and a score below the median score was categorized as low DRNK. The median score was used in this study because the Shapiro-Wilk test indicated that was not normally distributed (W (142) = 0.910, p<0.001).

## Data collection tool for the qualitative phase

Semi-structured interviews were conducted in English and Asante Twi (a Ghanaian vernacular) to explore the facilitators and barriers to dietary adherence. The questions asked included: the factors that enabled them to follow the dietary recommendations given by healthcare professionals and the factors that made it challenging for them to adhere to dietary recommendations received from healthcare professionals. The questions were developed based on the research objectives, a review of relevant literature, and feedback from experts and pre-testing with five persons with T2D at the Kumasi South Hospital in the Asokwa Municipal, Ashanti Region.

## Data collection procedure for the quantitative phase

Persons attending the outpatient diabetes clinics in the various hospitals were approached and the objectives of the study and procedures involved were explained to them. Persons who agreed to take part in the study gave written consent and the questionnaires were self-administered or interviewer-administered depending on the participant's choice. The completion of the questionnaires took about 10–15 minutes.

## Data collection procedure for the qualitative phase

Fourteen participants, selected from the participants who took part in the quantitative phase, were purposively sampled for individual face-to-face interviews after they had provided informed written consent. The interviews were audio-recorded and were conducted in English or Asante Twi depending on the participant's language of preference. The interviews lasted between 15 and 30 minutes and were conducted in a quiet location within the hospital premises to allow participants to freely express their views. Follow-up questions and probes were used to help gain more understanding of participants' responses. Data saturation was reached after conducting 14 interviews.

Interviews were facilitated by the first, third and fourth authors (either DW, RM or PA) whereby one led the discussions whereas the others took notes during the sessions. RM is a male whereas DW and PA are females; all three researchers have a background in nursing and have received training in the conduct of qualitative interviews and data analysis.

## Quantitative data processing and analysis

Data were analysed using Statistical Product and Service Solutions (SPSS) version 29.0 for Windows (IBM SPSS Statistics). There were no missing data in the survey responses as the researchers thoroughly examined all sections to ensure their completeness. Descriptive statistics such as means, medians, standard deviations, range, and percentages were used to describe continuous variables including age and diabetes-related nutrition knowledge. Categorical variables were presented in frequencies and percentages and data on dietary adherence were described using medians and interquartile ranges. To identify the factors associated with dietary adherence, a binary logistic regression analysis was done. The variables entered into the

binary logistic regression model were selected based on evidence from previous literature. Previous studies in Ghana and other countries have reported that age, sex, marital status, educational level, employment, monthly income, family history of diabetes, presence of comorbidities, duration of condition and dietary knowledge significantly influence dietary adherence [8, 12–14, 21, 41]. Both univariable and multivariable logistic regression analyses were done and reported using odds ratio and their 95% confidence intervals. The significance level was set at a p-value of $\leq 0.05$.

## Qualitative data analysis

After the end of each interview, member checking was done to allow participants to make the necessary clarifications or corrections to the main issues raised during the session. Of the 14 interviews, 11 of them were conducted in the local Ghanaian language (Asante Twi) whereas the remaining three were conducted using the English language. The interviews that were conducted in Twi were forward-translated into English and back-translated into the local language to ensure the accuracy of the translation process. The six-step framework for thematic analysis provided by Braun & Clark [42] was applied by the researchers during the analysis. The researchers involved in the data collection (DW, RM, and PA) independently read the transcripts several times for familiarisation, then developed and assigned codes to the data using NVivo software, version 20.0. The assignment of codes was done by identifying and labelling meaningful units of the data based on their relationship to the research questions. All codes were solely derived from the data. The three researchers then met to discuss generated codes and resolve discrepancies until a consensus was reached. Where required, disagreements were resolved through the second, fifth and eleventh authors (AKA-D, PAD and JK-D). The codes were later merged based on patterns and relationships between codes to form larger meaningful units (sub-themes and themes). The themes initially developed in the process of analysis were reviewed and refined by the first, second, third, fourth, fifth and eleventh authors (DW, AKA-D, RM, PA, PAD and JK-D) to ensure their consistency and relevance to research objectives. The final step involved naming the refined sub-themes and themes.

## Integration methods

At the methods level, integration was accomplished through **connecting** (explained in data collection for qualitative phase), while **joint display** was used at the interpretation and reporting level as described by Fetters et al. [35]. For the joint display, related quantitative and qualitative findings are brought together using visual elements such as figures, tables, matrices or graphs. According to Fetters et al. [35], the coherence of the quantitative and qualitative findings, known as the "fit" of data integration may be interpreted in three ways. The findings of the qualitative and quantitative findings may either validate each other (confirmation), contradict or disagree with each other (discordance) or both datasets may provide insights of the phenomenon of interest by addressing different aspects of the phenomenon (expansion) [35].

## Operational definitions

The independent variables were the sociodemographic variables (age, sex, marital status, place of residence, educational status, employment status and monthly income), clinical variables (duration of condition, family history of diabetes and presence of comorbidities) and diabetes related-nutritional knowledge.

**Diabetes-related nutritional knowledge** refers to participant's knowledge on recommended number of fruits and vegetables to be consumed daily, oils to cut down, foods to cut down a lot or increase intake, foods high or low in fibre, foods high or low in starch, foods

high or low in added sugar, foods with high glycaemic index, foods likely to raise blood cholesterol and healthy food choices.

The dependent variable was **dietary adherence.** Using the PDAQ, participants were classified as having good dietary adherence if they obtained a score equal to or above the median score (the median score in this study was 37.00).

## Quality assurance: Validity, reliability, and trustworthiness

The validity and reliability of the PDAQ and GNKQ-R have been established in previous studies [39, 40]. The PDAQ and GNKQ-R had Cronbach Alpha coefficients of 0.78 and 0.93 respectively in the previous studies. The PDAQ has been used in diabetes management research in Ghana [14] and Ethiopia [12, 13]. In the current study, the face and content validity of the data collection tool was established by a panel of six experts in diabetes management, nutrition, nursing, and instrument development. Following the validity test, both questionnaires were pretested among 20 persons with T2D who were receiving care (outpatient care) at the Kumasi South Hospital in the Asokwa Municipal, Ashanti Region, before the commencement of data collection. The Cronbach Alpha obtained for the PDAQ and GNKQ-R from the pre-test were 0.74 and 0.92 respectively. Following the pre-test, necessary amendments were made to the instructions of the questionnaire, contents, order of the questions, and grammatical issues based on findings from the pre-test.

The content of the interview guide was also evaluated by the six-member panel who reviewed the quantitative tools and pre-tested among 5 participants. The evaluation focused on the following areas: clarity of the content, appropriateness of wording, and terminology for the target population, grouping of questions, relevance of questions to study objectives and number of questions. The interview guide was then modified based on the comments of reviewers and findings from the pre-test.

Trustworthiness was ensured in the present study by applying the principles of credibility, conformability, dependability, and transferability [43, 44]. To ensure credibility, researcher triangulation was done during interviews and analysis as well as respondent validation after interviews. The detailed description of the research methods and findings also contributed to the dependability of the study. Additionally, transferability was established by providing a dense description of the demographic characteristics and geographic boundaries of the population in this study.

## Ethical considerations

Before the conduct of the study, permission was sought from the management of the two hospitals. Ethical approval with reference number CHRPE/387/22 (initial approval) and CHRPE/AP/588/22(protocol amendment) was obtained from the Committee on Human Research, Publication and Ethics (CHRPE) of the School of Medicine and Dentistry, Kwame Nkrumah University of Science and Technology (KNUST), Kumasi, Ghana. Ethical principles such as informed consent, confidentiality, autonomy, beneficence, non-maleficence, justice, and voluntary participation were adhered to throughout the study.

## Results

### Quantitative results

**Sociodemographic characteristics and clinical data.**   Between September and October 2022, a total of 142 persons participated in the survey out of 146 people who were approached yielding a response rate of 97.3%. Reasons given for non-participation included lack of time

**Table 1. Sociodemographic characteristics and clinical data of participants.**

| Variable | Frequency | Percentage | Mean (SD) |
|---|---|---|---|
| Age(years) | - | - | 60.74 (11.03) |
| *Sex* | | | |
| Male | 28 | 19.7 | - |
| Female | 114 | 80.3 | - |
| *Marital status* | | | |
| Married | 83 | 58.5 | - |
| Unmarried | 59 | 41.5 | - |
| *Place of residence* | | | |
| Urban | 120 | 84.5 | - |
| Rural | 22 | 15.5 | - |
| *Educational status* | | | |
| Educated (Formal education) α | 110 | 77.5 | - |
| Uneducated (No formal education) | 32 | 22.5 | - |
| *Employment status* | | | |
| Employed | 60 | 42.3 | - |
| Not employed β | 82 | 57.7 | - |
| *Monthly income (in Ghana Cedis)* | | | |
| <500 | 88 | 62.0 | - |
| ≥ 500 | 54 | 38.0 | - |
| *Family history of type 2 diabetes* | | | |
| Yes | 97 | 68.3 | - |
| No | 45 | 31.7 | - |
| *Presence of comorbidity* | | | |
| Yes | 118 | 83.1 | - |
| No | 24 | 16.9 | - |
| *Duration of condition* | | | |
| Less than 5 years | 52 | 36.6 | - |
| 5 years and above | 90 | 63.4 | - |

α-Primary education, secondary education, and tertiary education; β- includes retirees and unemployed participants.

(n = 3) and disinterest in the study (n = 1). Most of the participants (80.3%, n = 114) were females, were living in urban areas (84.5%, n = 120), and received a monthly income of less than 500 Ghana Cedis (62%, n = 88). More than half of them were married (58.5%, n = 83) and not employed (57%, n = 82) and about seventy-seven percent (77.5%, n = 110) of them were educated.

Regarding clinical data, a greater number of the participants had comorbidities (83.1%, n = 118). About two-thirds (68.3%, n = 97) of them had a family history of diabetes and sixty-three percent (63.4%, n = 90) of the participants had T2D for at least five years (**Table 1**).

**Diabetes-related nutrition knowledge.** Participants had a wide range of diabetes-related nutrition knowledge with scores ranging from 4(14.8%) to 25(92.6%) and a median score of 18 (66.7%). Slightly over half of the participants (53.5%, n = 76) in this study scored equal to or above the median score (indicating high diabetes-related nutritional knowledge).

**Dietary adherence.** About half (50.7%) of the participants had good dietary adherence (obtained a score equal to or above the median score).

The highest median (IQR) score [5.00 (3.00)] was recorded for the question on the intake of fish or other foods high in omega 3 and the lowest median (IQR) score was recorded for the

Table 2. Perceived Dietary Adherence Questionnaire (PDAQ) score for participants.

| Item | Median | Interquartile Range (IQR) |
|---|---|---|
| On how many of the last SEVEN DAYS have you followed dietary recommendations given by your healthcare provider? | 3.00 | 1.00 |
| On how many of the last SEVEN DAYS did you eat the number of fruit and vegetables you are supposed to eat based on recommendations from your healthcare provider? | 3.00 | 2.00 |
| On how many of the last SEVEN DAYS did you eat carbohydrate-containing foods with a low glycaemic index? (Example: beans, pasta, low-fat dairy products) | 3.00 | 3.00 |
| On how many of the last SEVEN DAYS did you eat foods high in sugar, such as cakes, cookies, desserts, candies, etc.? | 0.00 | 0.00 |
| On how many of the last SEVEN DAYS did you eat foods high in fibre such as oatmeal, high fibre cereals (wheat, brown rice), and whole-grain bread? | 3.00 | 4.00 |
| On how many of the last SEVEN DAYS did you space carbohydrates evenly throughout the day? | 4.00 | 4.00 |
| On how many of the last SEVEN DAYS did you eat fish or other foods high in omega-3 fats? | 5.00 | 3.00 |
| On how many of the last SEVEN DAYS did you eat foods that contained or were prepared with sunflower, soybean, or olive oils? | 2.00 | 3.00 |
| On how many of the last SEVEN DAYS did you eat foods high in fat (such as high-fat dairy products, fatty meat, fried foods, or deep-fried foods)? | 0.00 | 1.00 |
| **Overall adherence** | **Frequency** | **Percentage (%)** |
| Good | 72 | 50.7 |
| Poor | 70 | 49.3 |

question on the intake of foods high in sugar [0.00 (0.00)] and foods high in fat [0.00 (1.00)] (**Table 2**).

**Association between dietary adherence and sociodemographic data, clinical data, and diabetes-related nutrition knowledge (DRNK).** **Table 3** shows the results of the binary logistic regression. The results of the univariable logistic regression indicated that living in an urban area (COR = 3.259, 95% CI: 1.193–8.902, p = 0.021), being educated (COR = 2.377, 95% CI: 1.046–5.400, p = 0.039), receiving a monthly income more than or equal to 500 Ghana Cedis (COR = 2.539, 95% CI: 1.260–5.117, p = 0.009) and increase in DRNK (COR = 1.117, 95% CI: 1.027–1.216, p = 0.010) were associated with good dietary adherence.

In multivariable logistic regression, it was found that for every one-unit increase in DRNK score, the likelihood of a participant having good dietary adherence increased by approximately 10% (AOR = 1.101, 95% CI: 1.004–1.208, p = 0.041) (**Table 3**). Also, participants who were living in urban areas were about 3 times more likely to demonstrate good dietary adherence (AOR = 3.058, 95% CI: 1.014–9.221, p = 0.047).

### Qualitative results

**Sociodemographic characteristics of participants.** In all, 14 participants were purposively sampled from both hospitals. About half (57.1%, n = 8) of them were selected from the KNUST Hospital and were within the age range of 51–62 (50.0%, n = 7). Many of them were females (78.6%, n = 11), were married (78.6%, n = 11) and lived in urban areas (71.4%, n = 10). Fifty-seven percent (57.1%, n = 8) of them received a monthly income of less than 500 Cedis and about forty-two percent (42.9%, n = 6) were not formally educated. Nearly thirty-six percent (35.7%, n = 5) of them were not employed (**Table 4**).

**Facilitators of dietary adherence.** Four major themes were actively derived from the data: (1) access to information on diet, (2) individual food preferences and eating habits (3) perceived benefits of dietary adherence, and (4) the presence of social support.

*Access to information on diet.* According to participants, exposure to information on diabetes and diet served as a facilitating factor for adherence to dietary recommendations.

**Table 3. Association between dietary adherence and socio-demographic factors, clinical factors, and diabetes-related nutrition knowledge.**

| Variable | Dietary adherence | | COR (95% CI) | p-value | AOR (95% CI) | p-value |
|---|---|---|---|---|---|---|
| | Good | Poor (ref) | | | | |
| Age (years) | - | - | 1.006 (0.976–1.036) | 0.717 | 1.038 (0.996–1.081) | 0.080 |
| *Sex* | | | | | | |
| Male (ref) | 16 | 12 | | | | |
| Female | 56 | 58 | 0.724 (0.315–1.667) | 0.448 | 1.574 (0.533–4.647) | 0.412 |
| *Marital status* | | | | | | |
| Married | 46 | 37 | 1.578 (0.806–3.090) | 0.183 | 1.622 (0.701–3.753) | 0.258 |
| Unmarried (ref) | 26 | 33 | | | | |
| *Place of residence* | | | | | | |
| Rural (ref) | 6 | 16 | | | | |
| Urban | 62 | 58 | 3.259(1.193–8.902) | 0.021* | 3.058 (1.014–9.221) | 0.047* |
| *Educational status* | | | | | | |
| Educated | 61 | 49 | 2.377 (1.046–5.400) | 0.039* | 1.426(0.568–3.583) | 0.450 |
| Uneducated (ref) | 11 | 21 | | | | |
| *Employment status* | | | | | | |
| Employed | 36 | 24 | 1.917 (0.975–3.768) | 0.059 | 1.881 (0.800–4.422) | 0.147 |
| Not employed (ref) | 36 | 46 | | | | |
| *Monthly income (in Ghana Cedis)* | | | | | | |
| < 500 (ref) | 37 | 51 | | | | |
| ≥ 500 | 35 | 51 | 2.539 (1.260–5.117) | 0.009* | 2.030(0.862–4.783) | 0.105 |
| **Clinical data** | | | | | | |
| *Family history of type 2 diabetes* | | | | | | |
| Yes | 54 | 43 | 1.884 (0.918–3.864) | 0.084 | 2.147 (0.953–4.836) | 0.065 |
| No (ref) | 18 | 27 | | | | |
| *Presence of comorbidity* | | | | | | |
| Yes | 57 | 61 | 0.561 (0.228–1.382) | 0.209 | 0.496 (0.165–1.490) | 0.211 |
| No (ref) | 15 | 9 | | | | |
| *Duration of condition* | | | | | | |
| Less than 5 years (ref) | 24 | 28 | | | | |
| 5 years and above | 48 | 42 | 1.333 (0.672–2.644) | 0.410 | 1.300(0.539–3.134) | 0.559 |
| Diabetes-related nutrition knowledge | - | - | 1.117(1.027–1.216) | 0.010* | 1.101(1.004–1.208) | 0.041* |

Abbreviations and symbols: *, Statistically significant variables (The significance level was set at ≤ 0.05); COR, Crude odds ratio; AOR, Adjusted odds ratio; (ref), reference category.

Participants reported that dietary education received from nurses at the hospital enabled them to adhere to dietary recommendations. They also narrated how they read about diabetes on the internet and made efforts to put the information obtained into practice.

*Eee. . . what makes me adhere to them is that when we come to the hospital, we are given education on diet. There's this nurse-an elderly woman—who teaches us and I'm able to adhere to the recommendations well. . . even if you forget or you don't know but you come and from the education you receive here you realize that; "oh, these things are not good for my body", then you can decide to practice what you heard.* (I1, KNUST Hospital)

*Oh, I read. I read a lot; I read a lot about diabetes, actually. So anytime I get some new education I try to implement it. I read a lot on the internet.* (I4, KNUST Hospital)

**Table 4. Sociodemographic characteristics of participants for the qualitative phase.**

| Variable | Number | Percentage (%) |
|---|---|---|
| **Age** | | |
| 39–50 | 5 | 35.7 |
| 51–62 | 7 | 50.0 |
| 63–74 | 2 | 14.3 |
| **Gender** | | |
| Female | 11 | 78.6 |
| Male | 3 | 21.4 |
| **Place of residence** | | |
| Rural | 4 | 28.6 |
| Urban | 10 | 71.4 |
| **Marital status** | | |
| Single | 1 | 7.1 |
| Married | 11 | 78.6 |
| Divorced | 2 | 14.3 |
| **Level of education** | | |
| Primary education | 3 | 21.4 |
| Secondary education | 2 | 14.3 |
| Tertiary education | 3 | 21.4 |
| None | 6 | 42.9 |
| **Employment status** | | |
| Private sector | 2 | 14.3 |
| Public sector | 2 | 14.3 |
| Self-employment | 5 | 35.7 |
| Not employed | 5 | 35.7 |
| **Monthly income** | | |
| <500 | 8 | 57.1 |
| 500–900 | 1 | 7.1 |
| 1000–2000 | 3 | 21.4 |
| >2000 | 2 | 14.3 |
| **Hospital** | | |
| KNUST Hospital | 8 | 57.1 |
| Ejisu Government Hospital | 6 | 42.9 |

Dietary education provided in the hospital and other sources (e.g. internet) improved participants' understanding of the condition. Participants who came to an understanding of the chronic nature of T2D and accepted the condition reported that they were putting in efforts to self-manage the condition.

*Now I've realized it has come to stay so I'm doing my best to take care of myself.* (I12, Ejisu Government Hospital)

*Individual food preferences and eating habits.* The food preferences and eating habits of participants influenced their ability to adhere to dietary recommendations. Participants readily adapted to the diet regimen provided by healthcare professionals when they included their preferred food choices. Some participants preferred to eat homemade foods and were circumspect about the foods they chose to eat. They indicated that this attitude helped them to adhere to dietary recommendations. Those who had the habit of eating homemade foods also reported that it enabled them to make healthier food choices.

*For vegetable salad I really like eating it; even before I was diagnosed with this condition, I was eating them often.* (I3, KNUST Hospital)

*Yes. You know. . . I watch what I eat. I don't eat outside, most of the time. That was why I was telling you that sometimes if I eat in the morning and I'm not in the house, I find it very difficult to eat anything. I don't eat till I come to the house. So, one; I don't eat outside and two; I watch what I eat also. It's not everything that you can get hold of that you're supposed to put in your mouth; you have to check your health. I'm very careful about what I eat.* (I5, KNUST Hospital)

*Perceived benefits of dietary adherence.* The benefits of dietary adherence as perceived by participants and how these perceptions influenced participants' adherence to dietary recommendations are represented by this theme. Participants reported better glycaemic control in response to following dietary advice from healthcare professionals. They highlighted that they adhered to dietary recommendations to prevent complications and hospitalization. The desire for a cure was also indicated as a reason for adherence.

*To be very frank, I came to realize that when I keep on with the diet, it helps me because if I'm strictly on diet my sugar goes down than if I'm not.* (I8, KNUST Hospital)

*I was admitted once because of this condition so I take good care of myself, so I'm not admitted here again.* (I11, Ejisu Government Hospital)

*When you listen to the education then you adhere to it. You do what you are told to and you avoid what you are told to stay away from. That's how I see it. Because I want to be free from the condition, I follow the dietary advice. Because I want to be free from the condition, I follow the dietary advice* (I2, KNUST Hospital)

Those who were advanced in age and lived with one or a few family members adhered to dietary recommendations to avoid critical illness.

*I stay with my husband; it's just the two of us. All the children have moved out—some are working, and others are abroad. So, if I fail to follow the recommendations and one of us gets critically ill, who will help the other*? (I12, Ejisu Government Hospital)

*Presence of social support.* This theme described the diverse support that participants received from their families and close relations. According to the participants, their family members and close relations served as checks when they were tempted to eat "unhealthy" foods, helped when they faced financial challenges, and provided education on a healthy diet. Some participants also narrated how their family members decided to follow the diet recommended for them.

*I have a daughter I stay with currently; she is 21 years old. Sometimes she tells me; "Ma, it's not good for you to eat this". So, she helps me- she educates me on foods that can help me live longer.* (I13, Ejisu Government Hospital)

*I have children and I work too- I sell yam. So sometimes when I face financial problems, I call my children to send me money. My daughter is also a nurse, and she educates me about my diet.* (I11, Ejisu Government Hospital)

*I stay with my husband; it's just the two of us. . . So, whatever I don't eat, he also doesn't eat.* (I12, Ejisu Government Hospital)

**Barriers to dietary adherence.** Four barriers to dietary adherence were actively generated from the data: (1) inability to afford recommended diets, (2) barriers related to foods available in the environment, (3) barriers related to the social environment, and (4) conflict between dietary recommendations and individual food preferences and personal eating habits.

*Inability to afford recommended diets*. Issues related to finances were a predominant point that was highlighted as a challenge faced by participants. According to the participants, it was sometimes difficult to provide meals for the family, not to mention purchase of fruits. They had to eat the same meals as other family members because they could not afford the cost of providing separate meals for themselves and their dependents. Some participants failed to adhere to dietary recommendations because their sources of financial support were no longer available.

> *. . . as for that one I can't afford it, it is difficult to get money. Sometimes you want to buy fruits, but you don't have money. It's the same at home-sometimes it becomes difficult to even get money to purchase foodstuffs for a meal. All these are worrying, and the condition continues to worsen. . .I eat the same food as my children; I provide the money for meals at home, and I can't afford to cook separate meals for myself and them as well.* (I2, KNUST Hospital)

> *Yeah, definitely. . . I'm a family man and you know the economic situation. Sometimes you can't decouple your diet from that of your children. So usually, sometimes I end up eating what the children eat because I can't afford to provide them with their meals and mine.* (I4, KNUST Hospital)

> *I used to adhere to them but now due to financial constraints, I'm unable to do that. . .During those times, my children were doing well in business, so they supported me but now they are facing some financial problems, hence I'm unable to follow the recommendations.* (I7, Ejisu Government Hospital)

*Barriers related to foods available in the environment*. This theme describes how the types of foods available in the environment influence participant's adherence to dietary recommendations. For participants who travelled for trading, they had to resort to foods other than the recommended ones due to limited availability. Participants reported that they were compelled by hunger to eat other foods when there was limited availability of recommended foods.

> *It is difficult because sometimes the foods you are supposed to eat are not available. For example, with bread, we are told to eat wheat bread but sometimes it is not available so you can't purchase it. And it's difficult to stay hungry so you get a slice of the white bread and eat.* (I6, KNUST Hospital)

> *I go on trips to trade, and sometimes the foods I want to eat are not available there, so I buy maybe malt to drink.* (I11, Ejisu Government Hospital)

The lack of time also influenced food availability. Participants reported that they could not prepare meals due to busy work schedules.

> *Sometimes you are in the market, and you've not even been able to achieve your goals for the day; you haven't even bought foodstuffs to prepare food. Sometimes you close from work and it's past 5 pm and you are going home to prepare some food. The diet is really a problem.* (I14, KNUST Hospital)

*Barriers related to the social environment*. One aspect of the social environment that was highlighted as a barrier to dietary adherence was social gatherings. Participants described how they were tempted to eat "unhealthy foods" at social gatherings and some coping mechanisms they used. They packaged the foods and sent them home to their children or ate them in moderation.

> *That's what I'm saying that I try my best. Mmm. . .sometimes it is very tempting, but I try my best not to eat too much of what I'm not supposed to eat. If possible, I package the food and send it home to my kids and even if I eat, I eat moderately.* (I4, KNUST Hospital)

> *. . .even when I go to weddings, I don't eat the food served. I bring the food to my kids at home.* (I3, KNUST Hospital)

*Conflict between dietary recommendations and individual food preferences and eating habits.* Participants found it difficult to change their old dietary habits and adapt to the regimen provided by healthcare professionals. They expressed their concern about having to eat within specific periods and they also felt they could not reach their satiety level when they followed the required portion control. Some participants felt that "unhealthy foods" had good taste and were within easy reach.

> *As for the fruits, I don't eat them often because I don't really like them. If not for this condition, I'm not someone who usually buys fruits to eat.* (I3, KNUST Hospital)

> *I do my best but not every day. At first, you could eat any food at any time you wanted to but now you have been given specific times to eat- you don't have to eat after 7 pm. . . I'm also not able to eat to my satisfaction because I've been told to reduce the amount of food I eat. I have to enjoy sometimes.* (I14, KNUST Hospital)

> *Ah, it (unhealthy foods) tastes good. . . The food tastes good so you will eat it. And it's readily available too.* (I12, Ejisu Government Hospital)

## Mixed methods results

**Table 5** shows the integration of quantitative and qualitative results using the joint display method [35]. The quantitative results speak to the strength of the association between dietary adherence and independent variables such as place of residence, DRNK, educational status and monthly income while participant quotes from the qualitative data provide insight about the nature of the associations. However, in the case of marital status, the qualitative and quantitative findings are incongruous with each other. While the qualitative findings indicate that spousal support facilitates adherence, the results of the logistic regression indicate that no statistically significant association exists between marital status and dietary adherence.

## Discussion

This study aimed to determine the factors associated with adherence and explore the barriers and facilitators factors of dietary adherence among persons with T2D. Our study revealed that half of the participants demonstrated good dietary adherence. This finding is in contrast to previous studies that reported lower levels of good dietary adherence among participants in Yemen [8] and Ethiopia [12, 13]. The disparities in the findings may be explained by differences in the study settings, sociodemographic characteristics of study participants, as well as the types of foods in the various areas [12].

**Table 5. Mixed methods results using joint display.**

| Factor | Quantitative | Qualitative | | Fit of integration |
|---|---|---|---|---|
| | | **Joint display for merging quantitative (logistic regression) and qualitative results** | | |
| | | Theme | Participant quote | |
| Diabetes-related nutritional knowledge (DRNK) | In both univariable and multivariable logistic regression, DRNK score was found to be associated with good dietary adherence. The results of the multivariable logistic regression indicated that for every one unit increase in DRNK score, the likelihood of a participant having good dietary adherence increased by approximately 10% (AOR = 1.101, 95% CI: 1.004–1.208, p = 0.041). | Access to information on diet | *. . .what makes me adhere to them is that when we come to the hospital, we are given education on diet. . .even if you forget or you don't know but you come and from the education you receive here you realize that; "oh, these things are not good for my body", then you can decide to practice what you heard.* (I1, KNUST Hospital) | Both quantitative and qualitative findings provide insight on the relationship between DRNK and dietary adherence. While the quantitative results speak to the strength of the association, qualitative results speak to the nature of the association by showing that exposure to dietary information enables people with T2D to make the right food choices. |
| Educational status | The results of the univariable logistic regression showed that educated participants were about two times more likely to have good dietary adherence (COR = 2.377, 95% CI: 1.046–5.400, p = 0.039) as compared to those who were not educated. | Access to information on diet | *Oh, I read. I read a lot; I read a lot about diabetes, actually. So anytime I get some new education I try to implement it. I read a lot on the internet.* (I4, KNUST Hospital) | Both quantitative and qualitative findings expand insights on the relationship between educational status and dietary adherence. The quantitative results provide information on the strength of the association, while the qualitative results describe how education influences dietary adherence. Educated patients can access diabetes-related information by reading from various sources such as the internet; providing them with a better understanding of the role of diet in diabetes and subsequently influencing their dietary habits. |
| Monthly income | The results of the univariable logistic regression showed that participants who received monthly income of $\geq$ 500 Cedis were about three times more likely to have good dietary adherence (COR = 2.539, 95% CI: 1.260–5.117, p = 0.009) as compared to those who received $\leq$ 500 Cedis. | Inability to afford recommended foods | *. . . as for that one I can't afford it, it is difficult to get money. Sometimes you want to buy fruits, but you don't have money. . . .* (I2, KNUST Hospital) | Findings from both data sources expand insights of the relationship between monthly income and dietary adherence. The quantitative results give information on the strength of the association and qualitative results explain how income influences adherence- individuals with low-income levels may find it difficult to purchase healthy but costly foods, leading to non-adherence. |
| Place of residence | Participants who lived in urban areas were more likely to have good dietary adherence as compared to those living in rural areas in both univariable and multivariable logistic regression analysis. The multivariable logistic regression showed that participants who were living in urban areas were about **three** times more likely to demonstrate good dietary adherence (AOR = 3.058, 95% CI: 1.014–9.221, p = 0.047). | Barriers related to foods available in the environment. | *It is difficult because sometimes the foods you are supposed to eat are not available. For example, with bread, we are told to eat wheat bread but sometimes it is not available so you can't purchase it. And it's difficult to stay hungry so you get a slice of the white bread and eat.* (I6, Ejisu Government Hospital) | Both quantitative and qualitative findings provide insight on the relationship between place of residence and dietary adherence. While the quantitative results give information on the strength of the association, qualitative results expand on the nature of the association by highlighting one way in which an individual's place of residence may influence dietary habits. It shows that participants living in rural areas may face challenges with adherence due to limited food choices as compared to urban areas where there are variety of foods due to availability of supermarkets and grocery stores. |
| Marital status | Marital status did not have a statistically significant association with dietary adherence in both univariable (COR = 1.578, 95% CI: 0.806–3.090, p = 0.183). and multivariable (AOR = 1.622, 95% CI: 0.701–3.753, p = 0.258) logistic regression analysis. | Presence of social support | *I stay with my husband; it's just the two of us. . . So, whatever I don't eat, he also doesn't eat.* (I12, Ejisu Government Hospital) | The quantitative and qualitative results disagree with each other. While the qualitative data indicates that spousal support facilitated adherence, no statistically significant association was found between marital status and dietary adherence. |

From the interviews, it was revealed that access to information on diet enabled participants to adhere to dietary recommendations. This is corroborated by the results of the multivariable logistic regression, which indicated that the likelihood of the occurrence of good dietary adherence increased with an increase in diabetes-related nutrition knowledge (DRNK). Participants who have higher DRNK may have better insight into the role of diet in the management of diabetes and this may influence their decision to adhere to dietary recommendations. This highlights the importance of providing continuous dietary education and counselling services to people with T2D. Several studies have shown that the possession of adequate dietary knowledge significantly influences effective dietary practices. Research conducted by Jemal et al. [45] indicated that higher knowledge significantly resulted in improved dietary practice, supporting a similar finding in Ghana [46] which demonstrated a significant association between good nutritional knowledge and dietary adherence. Additionally, Adam et al. [34] conducted a cross-sectional study in Sudan among patients with T2D to assess dietary knowledge, attitude and practices. Adam et al. reported that most of the participants demonstrated good dietary knowledge which was evident in their dietary practice as nearly 58% of them demonstrated good dietary adherence. However, other studies [26, 47] did not find an association between nutrition knowledge and dietary practices of persons with T2D. In a study conducted by Han et al. [26] in Singapore, no correlation was found between DRNK and adherence to dietary guidelines. Another study conducted among patients with T2D in Thailand reported that there was no association between diabetes nutritional knowledge and actual dietary behaviour [47]. The differences in the knowledge levels and dietary practices of participants in this study and the previous studies may account for these observations.

Another factor that was found to have a statistically significant association with dietary adherence was place of residence. Similar to the findings of Alhariri et al. [8] in Yemen and that of Adam et al. [12] in Ethiopia, we found that participants living in urban areas were more likely to adhere to dietary recommendations compared to those living in rural areas. This observation may be attributed to the several factors related to differences between urban and rural areas. Firstly, people living in urban areas have access to a wide variety of foods due to the availability and easy access to supermarkets and grocery stores as compared to rural areas. For instance, in this study, participants living in rural areas mentioned that it was difficult to access recommended foods such as wheat bread. In such situations, they had to resort to white bread, which is less healthy, to satisfy their hunger. Secondly, patients in urban areas have better access to health facilities that provide dietary education. This can lead to greater awareness of the importance of diet and enable them to make better food choices. Finally, patients living in urban areas may have higher income levels because of better job opportunities. Patients with higher income will be able to afford a wide variety of foods and healthy foods.

Although DRNK and place of residence were the only variables found to be associated with dietary adherence based on the results of the multivariable logistic regression, the results from the qualitative phase of the study showed that dietary adherence may be influenced by multiple factors related to the individual and the environment. One predominant factor that has been reported in previous studies [15, 16] and also surfaced in this study was financial challenges. Participants highlighted the cost of foods, especially fruits, as a barrier to dietary adherence. Moreover, the sociodemographic data from the quantitative results showed that nearly 60% of the participants were unemployed and 62% received a monthly income of less than 500 Cedis. This challenge may be aggravated by the current economic hardship in Ghana. The depreciation of the Cedi and high inflation rates have increased the cost of living, particularly for foods [48]. Existing interventions such as the Livelihood Empowerment Against Poverty (LEAP) should be expanded to assist persons with diabetes. We also suggest that persons with T2D should take advantage of seasonal fruits which they prefer since they are cheaper.

Participants in this study and many others [11, 15–17, 23] have highlighted the social environment either as a facilitator or a barrier to dietary adherence. In this study, support from family members positively influenced dietary adherence. This implies that interventions that promote family engagement can be beneficial in improving diabetes management. An aspect of the social environment that was highlighted by participants as a barrier to dietary adherence was social events. Similarly, participants in a study conducted by Tewahido & Berhane [49] in Ethiopia reported that adherence during social events was difficult because refusing to eat a common dish during social events was unacceptable in their culture. In Ghana, social events such as weddings, naming ceremonies, and funerals form a significant part of the social and cultural norms and usually, the choice of foods available at such events may be less appropriate for persons with T2D. Persons with diabetes may be tempted to eat the foods offered because they feel left out as almost all event attendants eat and drink. Participants in this study dealt with this problem by packaging foods served at these events and sending them home to their children. In previous studies conducted in Accra (Ghana) [23] and Iowa (United States) [16] participants reduced or avoided interactions with family and friends that involved foods and drinks. One approach to dealing with this problem that was highlighted in a meta-ethnography study [50] is "strategic non-compliance"; whereby persons with diabetes strategically alter their diabetes regimen (diet restrictions, medications, testing, etc.) to allow for social events. Evidence from one of the studies in the meta-ethnography showed that persons who used this approach had better glucose control and it also improved the balance between quality of life and management of the condition [50].

Also, individual food preferences and eating habits either acted as a facilitator or barrier to dietary adherence. Some participants readily adapted to the dietary regimens because they included their preferred food choices while others found it difficult to include certain foods in their diet. Similar to findings in a previous study by Laranjo et al. in Portugal [15], some participants were concerned about sacrificing their satiety due to portion control and others felt the regime was stark in terms of time for eating. Given this, dietary education sessions and planning of dietary regimens should be individualized, taking into consideration individual food preferences and other personal dietary habits that could influence adherence.

Additionally, participants who lived in rural areas felt that their environment offered limited access to healthy food choices, and this deterred them from following dietary recommendations. Busy work schedules were another factor highlighted as a barrier to dietary adherence. This is supported by the findings of Mogre et al. [51], Adu et al. [52], Han et al. [26] and Bukhsh et al. [53]. This is of significant public health importance and hence policymakers should consider efforts to help make healthy foods readily available in communities.

In contrast with findings from a study in Accra [23], the fear of developing complications or getting critically ill compelled participants in this study to adhere to dietary recommendations. However, the finding concerning hope for a cure as a facilitator was in line with the previously mentioned study [23].

It was also revealed that understanding the chronic nature of diabetes and acceptance of diagnosis enabled participants to effectively participate in self-management practices. This implies that insight about diabetes and acceptance of diagnosis enabled participants to develop a positive attitude towards the management of their diabetes. This finding highlights the importance of education and counselling to help patients understand and manage diabetes, as well as accept their diagnosis.

## Implications for policy and healthcare delivery

The findings from this study imply that dietary adherence among persons with T2D is influenced by diabetes-related nutrition knowledge and multiple factors related to the individual

and the environment in which they live and work. Given this, health providers and other stakeholders interested in controlling diabetes should use multidimensional and individualized interventions to improve dietary adherence among persons with T2D. Healthcare professionals should provide continuous dietary education to persons with T2D. Additionally, dietary recommendations should be tailored to individual preferences and dietary habits. The Government of Ghana should expand poverty-reduction social interventions such as the Livelihood Empowerment Against Poverty (LEAP) to assist persons with T2D who face financial challenges. Lastly, policymakers should make efforts to improve access to healthy foods in communities.

## Limitations of the study

Patients who had a confirmed diagnosis less than one year, those who could not speak, understand, or write English/ Twi, and those who had sought care from the hospital less than twice in the past 12 months (using the study start date as reference) did not participate in this study. Additionally, critically ill patients and those with significant cognitive impairments were not included in this study. This may be a possible source of bias and may limit the generalization of findings to all persons living with T2D in Ghana. Also, the use of self-reporting in the evaluation of dietary adherence could have resulted in overestimation or underestimation due to social desirability bias. There is a tendency that the hospital environment influenced the participants to provide responses they felt were appropriate to investigators. Future research should focus on assessing dietary adherence outside the clinical setting. Despite these limitations, the findings of this study are relevant to improving dietary adherence among persons with T2D. Healthcare professionals should consider the individual preferences and situations of patients when providing care. Additionally, policymakers and other stakeholders should make efforts to develop policies and interventions that will ensure easy access to healthy foods and support patients facing financial difficulties.

## Supporting information

**S1 File.**
(DOCX)

**S2 File.**
(DOCX)

**S1 Data.**
(XLSX)

## Acknowledgments

We acknowledge the management and healthcare professionals in the various hospitals who provided support in this study. We also acknowledge all the participants in this study.

## Author Contributions

**Conceptualization:** Dorothy Wilson, Richard Marfo, Paulina Amoh, Precious Adade Duodu.

**Data curation:** Dorothy Wilson.

**Formal analysis:** Dorothy Wilson, Abigail Kusi-Amponsah Diji, Richard Marfo, Paulina Amoh, Precious Adade Duodu, Samuel Akyirem, Joana Kyei-Dompim.

**Investigation:** Dorothy Wilson, Richard Marfo, Paulina Amoh.

**Methodology:** Dorothy Wilson, Richard Marfo, Paulina Amoh, Precious Adade Duodu.

**Project administration:** Dorothy Wilson, Richard Marfo.

**Resources:** Dorothy Wilson, Abigail Kusi-Amponsah Diji, Richard Marfo, Paulina Amoh, Joana Kyei-Dompim.

**Supervision:** Abigail Kusi-Amponsah Diji, Joana Kyei-Dompim.

**Validation:** Abigail Kusi-Amponsah Diji, Precious Adade Duodu, Samuel Akyirem, Joana Kyei-Dompim.

**Visualization:** Dorothy Wilson.

**Writing – original draft:** Dorothy Wilson, Richard Marfo, Paulina Amoh.

**Writing – review & editing:** Dorothy Wilson, Abigail Kusi-Amponsah Diji, Richard Marfo, Precious Adade Duodu, Samuel Akyirem, Douglas Gyamfi, Hayford Asare, Jerry Armah, Nancy Innocentia Ebu Enyan, Joana Kyei-Dompim.

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
