## [Decision Letter · Decision Letter 0]

19 Sep 2023

PONE-D-23-08604Dietary adherence among persons with type 2 diabetes: A concurrent mixed methods studyPLOS ONE

Dear Dr. Wilson,

Thank you for submitting your manuscript to PLOS ONE. After careful consideration, we feel that it has merit but does not fully meet PLOS ONE’s publication criteria as it currently stands. Therefore, we invite you to submit a revised version of the manuscript that addresses the points raised during the review process.

Please submit your revised manuscript by Nov 03 2023 11:59PM. If you will need more time than this to complete your revisions, please reply to this message or contact the journal office at plosone@plos.org. Please include the following items when submitting your revised manuscript:A rebuttal letter that responds to each point raised by the academic editor and reviewer(s). You should upload this letter as a separate file labeled 'Response to Reviewers'.A marked-up copy of your manuscript that highlights changes made to the original version. You should upload this as a separate file labeled 'Revised Manuscript with Track Changes'.An unmarked version of your revised paper without tracked changes. You should upload this as a separate file labeled 'Manuscript'.

We look forward to receiving your revised manuscript.

Kind regards,

Engelbert A. Nonterah, MD, PhD

Academic Editor

PLOS ONE

Journal Requirements:

Reviewers' comments:

Reviewer's Responses to Questions

**Comments to the Author**

1. Is the manuscript technically sound, and do the data support the conclusions?

Reviewer #1: Partly

Reviewer #2: Yes

2. Has the statistical analysis been performed appropriately and rigorously? 

Reviewer #1: No

Reviewer #2: Yes

3. Have the authors made all data underlying the findings in their manuscript fully available?

Reviewer #1: Yes

Reviewer #2: Yes

4. Is the manuscript presented in an intelligible fashion and written in standard English?

Reviewer #1: No

Reviewer #2: Yes

5. Review Comments to the Author

Reviewer #1: The authors report their findings from a mixed methods study on adherence to dietary recommendations among persons living with diabetes in urban settings in Ghana . Their paper is generally well written with few language use errors which need to be corrected. Review of the paper by a native English speaker would be beneficial. The qualitative methods are described in explicit detail which adds to the methodological rigour. The methods, results and discussion are congruent however the study design chosen is inappropriate for the aim. A longitudinal design is required for determining predictors. At best a cross-sectional design may yield associations. The authors use p-values in deciding on the variable selection when modelling . Data-driven selection of variables is a methodological no go and known to result in spurious associations. Selection of variables should rather be based on theoretical knowledge and biological plausibility. The authors should consider this and re-run the qualitative analysis.

Finally the authors may find this reference useful : Lamptey R, Amoakoh-Coleman M, Djobalar B, Grobbee DE, Adjei GO, Klipstein-Grobusch K. Diabetes self-management education interventions and self-management in low-resource settings; a mixed methods study. Plos one. 2023 Jul 14;18(7):e0286974.

Reviewer #2: The study investigates dietary adherence among individuals with Type 2 diabetes in Ghana. Its primary objectives are to identify predictors of dietary adherence and explore the influencing factors. To achieve these goals, the study adopts a mixed-methods approach, combining quantitative surveys and qualitative interviews. The study's findings reveal that approximately half of the participants exhibit good dietary adherence. Various factors impact adherence, including diabetes-related nutrition knowledge, financial challenges, social environments, individual food preferences, and access to healthy food choices.

Suggestions for Improvement:

1. The authors should present in the introduction the main findings of studies done in Ghana with quantitative and qualitative data alone. This would provide important context for readers and help them understand the study's significance within the existing body of research.

2. The authors should explain why triangulation is a novel and superior approach compared to quantitative or qualitative studies alone, providing an example. This explanation can help readers understand the rationale behind the chosen research methodology and its advantages.

3. The authors should provide more detailed information about the sampling strategy, participant selection, randomization, and how the sample represents the target population.

4. The authors should offer insights into the characteristics of non-participants and discuss potential sources of bias introduced by non-participation, enhancing transparency in the study.

5. The authors should describe validity and reliability of the tools and instruments used for data collection, including survey instruments. This information helps readers assess validity of the study's findings.

6. The authors should offer a brief outline of the statistical methods used for analyzing quantitative data, specifying types of tests and significance levels for clarity.

7. The authors should provide a concise description of the qualitative data analysis process, including theme derivation, coding, and software tools used, ensuring transparency in qualitative analysis.

8. The authors should consider merging quantitative results from Tables 3 and 4 into a single regression analysis, addressing collinearity issues for streamlined presentation.

9. The authors should provide a more comprehensive and critical comparison of the study's findings with existing literature, exploring both similarities and differences and discussing implications for the field's understanding. This in-depth comparison helps readers contextualize the study's contributions and assess its significance in the broader research landscape.

6. PLOS authors have the option to publish the peer review history of their article (what does this mean?). If published, this will include your full peer review and any attached files.

Reviewer #1: **Yes: **Roberta Lamptey

Reviewer #2: **Yes: **Felix P Chilunga

---

## [Author Response · Author response to Decision Letter 0]

3 Nov 2023

ADDITIONAL EDITOR’S COMMENTS

COMMENT RESPONSE

1. Please ensure that your manuscript meets PLOS ONE's style requirements, including those for file naming. Thank you for the comment. We have prepared the manuscript based on the PLOS ONE style requirements.

2. Please include captions for your Supporting Information files at the end of your manuscript, and update any in-text citations to match accordingly. Thank you for the comment. We have included captions for Supporting Information files at the end of the manuscript. Please refer to page 52, lines 621 to 624. In-text citations have also been updated.

REVIEWER 1 COMMENTS

 The authors report their findings from a mixed methods study on adherence to dietary recommendations among persons living with diabetes in urban settings in Ghana . Their paper is generally well written with few language use errors which need to be corrected. Review of the paper by a native English speaker would be beneficial. Thank you for pointing this out. The manuscript has been reviewed by a native English speaker and the language errors have been corrected.

The methods, results and discussion are congruent however the study design chosen is inappropriate for the aim. A longitudinal design is required for determining predictors. At best a cross-sectional design may yield associations. Thank you for the comment. We agree with this comment. We have accordingly revised the study’s aim. The revised aim is to identify the factors associated with dietary adherence and explore the barriers and faciltators to dietary adherence. Please refer to page 2, lines 17 and 18 and page 8, line 107 and 108.

The authors use p-values in deciding on the variable selection when modelling. Data-driven selection of variables is a methodological no go and known to result in spurious associations. Selection of variables should rather be based on theoretical knowledge and biological plausibility. The authors should consider this and re-run the qualitative analysis. Thank you for your comments. Based on your recommendation we have re-run the quantitative analysis. The variables included in the model were selected based on evidence from previous literature conducted in Ghana and other countries; on the factars associated with dietary adherence. Please refer to page 14, lines 202 to 204 and to pages 24 to 27, Table 3 for the results of the analysis.

However, we believe that the qualitative analysis was well conducted and the results reflect the views of the participants. Additionally, your preceding comments are related to quantitative analysis rather than qualitative analysis. 

REVIEWER 2 COMMMENTS

1. The authors should present in the introduction the main findings of studies done in Ghana with quantitative and qualitative data alone. This would provide important context for readers and help them understand the study's significance within the existing body of research.

 Thank you for the suggestion. This has been included in the introduction. Please refer to pages 6 and 7, lines 78 to 92. 

2. The authors should explain why triangulation is a novel and superior approach compared to quantitative or qualitative studies alone, providing an example. This explanation can help readers understand the rationale behind the chosen research methodology and its advantages Thanks for the comment. This has been included in the introduction. Please refer to pages 7 and 8, line 92 to 106.

3. The authors should provide more detailed information about the sampling strategy, participant selection, randomization, and how the sample represents the target population Thanks for the comment. We have accordingly added more information to emphasize this point. For the quantitative sampling, we explained how we used consecutive sampling to select participants who met the inclusion criteria until the desired sample size was reached. Please refer to page 10, line 134. Also for the qualitative phase, we explained how researchers selected participants from the quantitative sample during data collection for the quantitative phase. We purposefully selected participants we believed would provide valuable information. These judgements were made based on researchers’ interaction with participants during the quantitative phase. Please refer to page 10, lines 137 to 141.

4. The authors should offer insights into the characteristics of non-participants and discuss potential sources of bias introduced by non-participation, enhancing transparency in the study.

. Thank you for pointing this out. We have incorporated your suggestion in the manuscript. The characteristics of non-participants based on the inclusion and exclusion criteria outlined in the manuscript have been provided in the limitations section of the manuscript. We added that the exclusion of patients with certain characteristics may be a source of bias and may limit the generalizability of the findings to all patients with type 2 diabetes in Ghana. However, we emphasized that the findings of the study are relevant for improving dietary adherence despite this limitation. Please refer to page 42, lines 486 to 490 and lines 493 to 497.

5. The authors should describe validity and reliability of the tools and instruments used for data collection, including survey instruments. This information helps readers assess validity of the study's findings Thank you for the comment. The validity and reliability of the tools used for quantitative data collection have been provided in the manuscript. The Cronbach Alpha of the questionnaires were reported as well as the process for establishing face and content validity. Please refer to page 16, lines 226 to 234.

Based on your comments, the process for establishing the content validity of the interview guide has been added. Please refer to page 16, lines 235 to 238.

6. The authors should offer a brief outline of the statistical methods used for analyzing quantitative data, specifying types of tests and significance levels for clarity.

 Thank you for the comment. The specific tests used for quantitative data analysis and the level of significance considered in the analysis have been reported in the manuscript. The reasons for using specific tests and the variables included were also outlined. The significance level was set at ≤0.05. Please refer to page 14, lines 199 to 210.

7. The authors should provide a concise description of the qualitative data analysis process, including theme derivation, coding, and software tools used, ensuring transparency in qualitative analysis Thank you for your suggestion. Details on the qualitative analysis have been added; proving information on the analysis process and the software tool used which is Nvivo software, version 20.0.

8. The authors should consider merging quantitative results from Tables 3 and 4 into a single regression analysis, addressing collinearity issues for streamlined presentation Thank you for the comment. The tables have been merged. The table contains results of the Spearman’s correlarion, chi-square test and binary logistic regression. Please refer to pages 24-27, Table 3.

9. The authors should provide a more comprehensive and critical comparison of the study's findings with existing literature, exploring both similarities and differences and discussing implications for the field's understanding. This in-depth comparison helps readers contextualize the study's contributions and assess its significance in the broader research landscape.

 Thank you for the comment. The discussion has been improved by critically comparing the study’s findings with others studies and establishing differences or similarities between quantitative and qualitative findings of the study. Please refer to pages of 38 and 39, lines 416 to 441; page 40, l

---

## [Decision Letter · Decision Letter 1]

16 Nov 2023

PONE-D-23-08604R1Dietary adherence among persons with type 2 diabetes: A concurrent mixed methods studyPLOS ONE

Dear Dr. Dorothy Wilson,

Thank you for submitting your manuscript to PLOS ONE. After careful consideration, we feel that it has merit but does not fully meet PLOS ONE’s publication criteria as it currently stands. Therefore, we invite you to submit a revised version of the manuscript that addresses the points raised during the review process.

**You have revised the manuscript according tp the review comments, however the manuscript could still benefit from additional revisions before it is deemed worthy of publication. Kindly address the following:**

**1. Correct the in-text reference in page 6 (Doglikuu et al 2021). Be consistent with the referencing style.**

**2. The  facilities sampled from have a combined T2DM attendance of 200, why did you have to calculate a sample size. The easiest will be to utilise the entire sample and do a retrospective power calculation to confirm whether you are adequately powered. The approach for the sample calculation problematic and that is the main point of the second reviewer.**

**3. Add a section to your methods where you define all the variables used. For instance how did you define rural and urban for patients attending two facilities within an urban municipality in the Ashanti region**

**4. you indicate media scores were used for you PDAQ and DRNK yet in table  2 you present mean and standard deviation. Correct this to median scores with IQR for clarity to the reader.**

**5. Summarise and discuss the overlap (triangulation) of quantitative and qualitative findings/results. You could illustrate this on put in a table.**

We look forward to receiving your revised manuscript.

Kind regards,

Engelbert A. Nonterah, MD, PhD

Academic Editor

PLOS ONE

Reviewers' comments:

Reviewer's Responses to Questions

**Comments to the Author**

1. If the authors have adequately addressed your comments raised in a previous round of review and you feel that this manuscript is now acceptable for publication, you may indicate that here to bypass the “Comments to the Author” section, enter your conflict of interest statement in the “Confidential to Editor” section, and submit your "Accept" recommendation.

Reviewer #1: (No Response)

Reviewer #2: All comments have been addressed

2. Is the manuscript technically sound, and do the data support the conclusions?

Reviewer #1: Partly

Reviewer #2: Yes

3. Has the statistical analysis been performed appropriately and rigorously? 

Reviewer #1: No

Reviewer #2: Yes

4. Have the authors made all data underlying the findings in their manuscript fully available?

Reviewer #1: (No Response)

Reviewer #2: Yes

5. Is the manuscript presented in an intelligible fashion and written in standard English?

Reviewer #1: Yes

Reviewer #2: Yes

6. Review Comments to the Author

Reviewer #1: The authors indicate a modification in the study aim to study associations rather than predictors. This is possible with a cross-sectional design but this must be an analytical cross-sectional design . The sample size formula used is for one sample with finite correction. This is a fundamental flaw.The investigators need to demonstrate that the sample size is adequate to be able to estimate the population parameter in this case OR

Reviewer #2: Thanks to the authors for addressing all my previous comments effectively. The manuscript has improved, and I have only a few additional suggestions for further enhancement:

1 It appears that the manuscript is currently in landscape layout. If this is the case, I recommend changing it to portrait format for better readability.

2 The introduction, while comprehensive, could benefit from being more concise for clarity.

3 Please provide the baseline N for the sample size calculation.

4 Include the justification for conducting the study at the two hospitals.

5 Clearly explain the rationale for selecting 14 participants and outline the criteria used to make this decision.

6 In the table presenting regressions, consider using the term "reference" instead of numerical values (1 or dashes) for clarity purposes.

7 Additionally, I suggest consolidating all statistically significant factors into one logistic regression model as predictors. Exclude collinear variables using variance inflation factors, and then report the final model, explicitly revealing the most important determinants.

7. PLOS authors have the option to publish the peer review history of their article (what does this mean?). If published, this will include your full peer review and any attached files.

Reviewer #1: **Yes: **Roberta Lamptey

Reviewer #2: No

---

## [Author Response · Author response to Decision Letter 1]

24 Jan 2024

Editor’s Comments

Comment 1: Correct the in-text reference in page 6 (Doglikuu et al 2021). Be consistent with the referencing style.

Response: Thank you for pointing this out. We have ensured that all references are in Vancouver style.

Comment 2: The facilities sampled from have a combined T2DM attendance of 200, why did you have to calculate a sample size. The easiest will be to utilize the entire sample and do a retrospective power calculation to confirm whether you are adequately powered. The approach for the sample calculation problematic and that is the main point of the second reviewer.

Response: Thank you for the comment. While we generally agree with the reviewer that recruiting the entire population of T2D individuals from the two facilities would have been ideal, we also understand that this was not feasible as some patients may decline participation or may not be reachable during the study period. The Yamane Taro approach to sample size calculation enabled us to determine the minimum number of participants that can be considered as representative of a finite population size of 200. That said, we have performed a retrospective power analysis with the 146-sample size and have provided details of this in the manuscript. Please refer to pages 9-10, lines 185 to 188.

Comment 3: Add a section to your methods where you define all the variables used. For instance, how did you define rural and urban for patients attending two facilities within an urban municipality in the Ashanti region.

Response: Thank you for the comment. A section for definition of independent and dependent variables has been included in the methods. Please refer to pages 16-17, lines 324 to 336.

However, the two health facilities are within two different municipalities in the Ashanti region. While the KNUST Hospital is in the Oforikrom municipal, which is an urban municipal, the Ejisu Government Hospital is in the Ejisu Municipal, which is made up of both urban and rural localities (based on the Ghana 2021 Population and Housing Census General Report Volume 3A: Population of Regions and Districts). This information has been added to the section for study setting in the methods and the names of the facilities have been included. Please refer to pages 7-8, lines 142 to 150. For the place of residence, the name of the locality of residence was obtained from the participants and were designated as rural or urban. All localities within the Oforikrom Municipality were designated as urban. For the Ejisu municipal, all localities apart from Ejisu (Krapa inclusive), Kwamo and Fumesua were designated as rural (based on information from the Ministry of Food and Agriculture website; https://mofa.gov.gh/site/sports/district-directorates/ashanti-region/162-ejisu-juaben and Openalfa Street Directories; https://ghana-streets.openalfa.com/ejisu-municipal-district). Please refer to page 11, lines 211 to 215.

Comment 4: You indicate media scores were used for you PDAQ and DRNK yet in table 2 you present mean and standard deviation. Correct this to median scores with IQR for clarity to the reader.

Response: Thank you for the comment. The mean and standard deviation in Table 2 have been changed to median and interquartile range respectively. Please refer to pages 21-23.

Comment 5: Summarize and discuss the overlap (triangulation) of quantitative and qualitative findings/results. You could illustrate this on put in a table.

Response: Thank you for the suggestion. The quantitative and qualitative findings have been integrated and presented in the results section using the joint display method as described by Fetters et al. in their article titled; “Achieving Integration in Mixed Methods Designs — Principles and Practices” (https://onlinelibrary.wiley.com/doi/10.1111/1475-6773.12117). Please refer to Table 5 on pages 36-43.

Reviewer 1 Comments

Comment 1: The authors indicate a modification in the study aim to study associations rather than predictors. This is possible with a cross-sectional design but this must be an analytical cross-sectional design. The sample size formula used is for one sample with finite correction. This is a fundamental flaw. The investigators need to demonstrate that the sample size is adequate to be able to estimate the population parameter in this case OR

Response: Thank you for the comment. We have indicated in the manuscript that the quantitative phase of the study employed an analytical cross-sectional design. Please refer to page 7, lines 126 to 127.Concerning the sample size, we have performed a retrospective power analysis using GPower and we determined that a sample size of 146 will yield >99% power at 95% confidence level. This determination was based on an assumed odds ratio of 4.7 between diabetes knowledge and dietary adherence as reported in a previous study in Sudan (reference provided in manuscript) and a two-tailed z-test. Please refer to page 9-10, lines 185 to 188.

Reviewer 2 Comments

Comment 1: It appears that the manuscript is currently in landscape layout. If this is the case, I recommend changing it to portrait format for better readability.

Response: Thank you for the suggestion. The manuscript layout has been changed to portrait.

Comment 2: The introduction, while comprehensive, could benefit from being more concise for clarity.

Response: Thank you for the suggestion. While we thought a comprehensive introduction could put the topic in context, we have attempted to make it a bit more concise. Please refer to pages 4-6.

Comment 3: Please provide the baseline N for the sample size calculation.

Response: Thank you for the comment. The baseline N for the sample size was 200 and it has been indicated in the manuscript. Please refer to page 9, lines 181-182.

Comment 4: Include the justification for conducting the study at the two hospitals.

Response: Thank you for the suggestion. The study sought to target both rural and urban population. Hence, the choice of the KNUST Hospital and Ejisu Government Hospital is based on their diverse patient populations. The KNUST Hospital provides services to a mix of students, university staff and locals while the Ejisu Government Hospital serves both rural and urban communities. Please refer to page 8, lines 156-162.

Comment 5: Clearly explain the rationale for selecting 14 participants and outline the criteria used to make this decision.

Response: Thank you for the comment. Data saturation (when new data replicates what has already been gathered) was achieved after conducting 14 interviews. The researchers selected participants who differ across multiple criteria such as age, sex, place of residence, income, education level, employment, and marital status. Please refer to page 10, lines 192 to 205.

Comment 6: In the table presenting regressions, consider using the term "reference" instead of numerical values (1 or dashes) for clarity purposes.

Response: Thank you for the suggestion. The numerical values (1 and dashes) have been deleted and the abbreviation ref (reference category), has been used to indicate categorical variables used as reference categories in the regression analyses. Please refer to Table 3, pages 23 to 26.

Comment 7: Additionally, I suggest consolidating all statistically significant factors into one logistic regression model as predictors. Exclude collinear variables using variance inflation factors, and then report the final model, explicitly revealing the most important determinants.

Response: Thank you for the comment. However, the primary purpose of the study is not to determine predictors of dietary adherence but to identify factors associated with dietary adherence. To determine predictors, a longitudinal study design will be required.

---

## [Editor Report · Decision Letter 2]

6 Feb 2024

PONE-D-23-08604R2Dietary adherence among persons with type 2 diabetes: A concurrent mixed methods studyPLOS ONE

Dear Dr. Wilson,

Thank you for submitting your manuscript to PLOS ONE. After careful consideration, we feel that it has merit but does not fully meet PLOS ONE’s publication criteria as it currently stands. Therefore, we invite you to submit a revised version of the manuscript that addresses the points raised during the review process.

We look forward to receiving your revised manuscript.

Kind regards,

Engelbert A. Nonterah, MD, PhD

Academic Editor

PLOS ONE

Journal Requirements:

**Additional Editor Comments:** Kindly address all comments from the two reviewersPlease address issues regarding the sample size pointed out by the two reviewersTo avoid further delays in arriving at a final decision, kindly return the revised manuscript in two weeks time (19th February, 2024)

---

## [Author Response · Author response to Decision Letter 2]

14 Mar 2024

EDITOR'S COMMENTS

Kindly address all comments from the two reviewers.

Response: All comments from both reviewers have been thoroughly addressed.

Comment: Please address issues regarding the sample size pointed out by the two reviewers.

Response: Concerning the issue on sample size raised by the first reviewer, we indicated that while we generally agree with the reviewer that recruiting the entire population of T2D individuals from the two facilities would have been ideal, we also understand that this was not feasible as some patients may decline participation or may not be reachable during the study period. The Yamane Taro approach to sample size calculation enabled us to determine the minimum number of participants that can be considered as representative of a finite population size of 200. That said, we performed a retrospective power analysis with the 146-sample size and have provided details of this in the manuscript. Please refer to page 9-10, lines 185 to 188.

The second reviewer’s comment was that we should provide the population size (baseline N) and we indicated that the population size was 200. Please refer to page 9, lines 181-182

REVIEWER 1 COMMENTS

Comment: The authors indicate a modification in the study aim to study associations rather than predictors. This is possible with a cross-sectional design but this must be an analytical cross-sectional design . The sample size formula used is for one sample with finite correction. This is a fundamental flaw. The investigators need to demonstrate that the sample size is adequate to be able to estimate the population parameter in this case OR 

Response: Thank you for the comment. We have indicated in the manuscript that the quantitative phase of the study employed an analytical cross-sectional design. Please refer to page 7, lines 126 to 127.Concerning the sample size, we have performed a retrospective power analysis using GPower and we determined that a sample size of 146 will yield >99% power at 95% confidence level. This determination was based on an assumed odds ratio of 4.7 between diabetes knowledge and dietary adherence as reported in a previous study in Sudan (reference provided in manuscript) and a two-tailed z-test. Please refer to page 9-10, lines 185 to 188.

REVIEWER 2 COMMENTS

Comment: It appears that the manuscript is currently in landscape layout. If this is the case, I recommend changing it to portrait format for better readability. 

Response: Thank you for the suggestion. The manuscript layout has been changed to portrait.

Comment: The introduction, while comprehensive, could benefit from being more concise for clarity. 

Response: Thank you for the suggestion. While we thought a comprehensive introduction could put the topic in context, we have attempted to make it a bit more concise. Please refer to pages 4-6

Comment: Please provide the baseline N for the sample size calculation 

Response: Thank you for the comment. The baseline N for the sample size was 200 and it has been indicated in the manuscript. Please refer to page 9, lines 181-182.

Comment: Include the justification for conducting the study at the two hospitals. 

Response: Thank you for the suggestion. The study sought to target both rural and urban population. Hence, the choice of the KNUST Hospital and Ejisu Government Hospital is based on their diverse patient populations. The KNUST Hospital provides services to a mix of students, university staff and locals while the Ejisu Government Hospital serves both rural and urban communities. Please refer to page 8, lines 156-162. 

Comment: Clearly explain the rationale for selecting 14 participants and outline the criteria used to make this decision. 

Response: Thank you for the comment. Data saturation (when new data replicates what has already been gathered) was achieved after conducting 14 interviews. The researchers selected participants who differ across multiple criteria such as age, sex, place of residence, income, education level, employment, and marital status. Please refer to page10, lines 192 to 205.

Comment: In the table presenting regressions, consider using the term "reference" instead of numerical values (1 or dashes) for clarity purposes. 

Response: Thank you for the suggestion. The numerical values (1 and dashes) have been deleted and the abbreviation ref (reference category), has been used to indicate categorical variables used as reference categories in the regression analyses. Please refer to Table 3, pages 23 to 26.

Comment: Additionally, I suggest consolidating all statistically significant factors into one logistic regression model as predictors. Exclude collinear variables using variance inflation factors, and then report the final model, explicitly revealing the most important determinants 

Response: Thank you for the comment. However, the primary purpose of the study is not to determine predictors of dietary adherence but to identify factors associated with dietary adherence. To determine predictors, a longitudinal study design will be required.

---

## [Editor Report · Decision Letter 3]

16 Apr 2024

Dietary adherence among persons with type 2 diabetes: A concurrent mixed methods study

PONE-D-23-08604R3

Dear Dr. Dorothy Wilson,

We’re pleased to inform you that your manuscript has been judged scientifically suitable for publication and will be formally accepted for publication once it meets all outstanding technical requirements.

Kind regards,

Engelbert A. Nonterah, MD, PhD

Academic Editor

PLOS ONE